# Oncostatin M Receptor Type II Knockout Mitigates Inflammation and Improves Survival from Sepsis in Mice

**DOI:** 10.3390/biomedicines11020483

**Published:** 2023-02-08

**Authors:** Saad Y. Salim, Nour AlMalki, Kimberly F. Macala, Alyssa Wiedemeyer, Thomas F. Mueller, Thomas A. Churchill, Stephane L. Bourque, Rachel G. Khadaroo

**Affiliations:** 1Department of Surgery, University of Alberta, Edmonton, AB T6G 2B7, Canada; 2Centre of Excellence for Gastrointestinal Inflammation and Immunity Research (CEGIIR), University of Alberta, Edmonton, AB T6G 2X8, Canada; 3Department of Critical Care Medicine, University of Alberta, Edmonton, AB T6G 2B7, Canada; 4Department of Anesthesiology and Pain Medicine, University of Alberta, Edmonton, AB T6G 2G3, Canada

**Keywords:** fecal slurry induced peritonitis, knockout mice, macrophage recruitment, inflammation, bacterial clearance

## Abstract

Sepsis remains one of the leading causes of death worldwide. Oncostatin M (OSM), an interleukin (IL)-6 family cytokine, can be found at high levels in septic patients. However, little is known about its role in sepsis. This study aimed to determine if the genetic knockout of OSM receptor (OSMR) type II signaling would improve survival in a murine model of sepsis. Aged (>50 weeks) OSMR type II knockout (KO) mice and wild-type (WT) littermates received an intraperitoneal injection of fecal slurry (FS) or vehicle. The KO mice had better survival 48 h after the injection of FS than the WT mice (*p* = 0.005). Eighteen hours post-FS injection, the KO mice had reduced peritoneal, serum, and tissue cytokine levels (including IL-1β, IL-6, TNFα, KG/GRO, and IL-10) compared to the WT mice (*p* < 0.001 for all). Flow cytometry revealed decreased recruitment of CD11b^+^ F4/80^+^ Ly6c^high+^ macrophages in the peritoneum of KO mice compared to WT mice (34 ± 6 vs. 4 ± 3%, P_Int_ = 0.005). Isolated peritoneal macrophages from aged KO mice had better live *E. coli* killing capacity than those from WT mice (*p* < 0.001). Peritoneal lavage revealed greater bacterial counts in KO mice than in WT mice (KO: 305 ± 22 vs. 116 ± 6 CFU (×10^9^)/mL; *p* < 0.001). In summary, deficiency in OSMR type II receptor signaling provided a survival benefit in the progression of sepsis. This coincided with reduced serum levels of pro-inflammatory (IL-1β, TNFα, and KC/GRO) and anti-inflammatory markers (IL-10), increased bacterial killing ability of macrophages, and reduced macrophage infiltration into to site of infection.

## 1. Introduction

Sepsis is a life-threatening organ dysfunction caused by a dysregulated host response to a pathogen. Despite notable advances in critical care, sepsis remains a leading cause of death in hospitalized patients [1,2,3], especially in the Intensive Care Unit [4,5]. Though individuals of all ages may develop sepsis and septic shock, the elderly (>65 years of age) account for a preponderance of affected patients [6]. Despite advanced age being an established risk factor for sepsis-related mortality, elderly patients are often excluded from both clinical and preclinical studies on the diagnosis and management of sepsis [7]. As the elderly population continues to grow, the use of aging models in biomedical research is becoming increasingly relevant; this is particularly true in the context of sepsis, since aging is known to have profound effects on virtually all systems and pathways implicated in its pathophysiology [6].

Cytokines are key mediators in the pathophysiology of sepsis and its progression to septic shock. Septic patients present increased levels of pro- and anti-inflammatory cytokines, including Tumor Necrosis Factor α (TNFα) and interleukin (IL)-1β, IL-6, IL-8, and IL-10 [7,8,9]. It has been proposed that the development of Multiple Organ Dysfunction Syndrome in sepsis is related to the amount of pro- and anti-inflammatory cytokines produced by the innate immune system, which mediates responses of the adaptive immune system [10]. While this dual-arm immune response is generally responsible for battling infection to restore homeostasis, uncontrolled and prolonged cytokine production may lead to a dysregulated inflammatory response culminating in shock, organ dysfunction, and death.

Oncostatin M (OSM) is a cytokine member of the IL-6 family [11]. Depending on the species, OSM can signal via two morphologically distinct receptor complexes [12]. OSM initially binds with low affinity to the alpha subunit gp130 and then heterodimerizes with high affinity with either the leukemia inhibitory factor receptor (LIFR) β subunit (gp130α/LIFRβ), forming the type 1 complex (OSMR type I), or the OSM receptor β subunit (gp130α/OSMRβ), forming the type II complex (OSMR type II) [13]. Though other cytokines (e.g., IL-6) interact with OSMR type I, it is thought that only OSM can bind to the OSMR type II complex. Notably, whereas human and rat OSM signals via both receptors complexes, murine OSM signals almost exclusively via OSMR type II [14]. As such, the OSMR^−/−^ mouse model is attractive for studying the role of OSMR type II in immune cell function during the acute-phase response [15]. OSM receptors are widely expressed in a variety of cells, including leukocytes, endothelial cells, hepatocytes, neurons, and some epithelial cells [16]. While OSM can elicit diverse immune responses, an important property is its ability to modulate the responses of other cytokines, where co-signaling with other cytokines can induce synergistic inflammatory responses [17]. Consequently, targeting the OSMR type II complex using antibody-based approaches has been effective in reducing inflammation and inhibiting leukocyte migration in rheumatoid arthritis [18], underscoring an important role of this receptor in immune function. OSM levels are elevated in patients with sepsis and septic shock [8,19], implicating OSM signaling in sepsis pathophysiology, although its role remains unclear. Here, we hypothesized that the abrogation of OSMR type II signaling could mitigate the inflammatory response during sepsis and in turn improve survival in mice. As noted above, aging is associated with greater susceptibility to sepsis and increased morbidity and mortality; therefore, the experimental design of this study included an experimental group of mice with advanced age to increase the relevance of the results to the clinical needs.

## 2. Materials and Methods

### 2.1. Animal Model

The protocols described herein were approved by the University of Alberta Animal Care and Use Committee (Animal Use Protocol No. 00000414) in accordance with the guidelines established by the Canadian Council on Animal Care. Male and female OSMR type II (OSMR^−/−^) knockout (KO) mice and wild-type (WT) littermates (OSMR^+/+^), bred on a C57BL/6 background were purchased from Charles River (St. Constant, QC, Canada). The mice were housed in rooms in the University of Alberta animal care facility, which maintained a 12 h light:12 h dark cycle and a room temperature of 22 ± 1 °C. The mice were housed in shoebox cages (5 mice/cage) containing nesting material, aspen chip bedding, and polyvinyl chloride tubing. The mice were given ad libitum access to a standard grain-based rodent chow (PicoLab 5LOD, LabDiet, St. Louis, MO, USA) and tap water. Animal care personnel performed welfare assessments routinely and prior to experimentation. The mice were used for experiments at approximately 12 weeks of age (range: 11–14 weeks; young group) or were aged in the animal care facility at the University of Alberta until 50–70 weeks of age (aged group). All live animal experiments took place in the same laboratory operating theatre setting during daytime hours.

### 2.2. Induction of Sepsis and Necropsy

A fecal-induced peritonitis (FIP) model of sepsis was used as previously described [20]. The cecal contents from male and female C57BL/6 mice (9–13 weeks of age) with no signs of infection were collected, weighed, and suspended in 5% dextrose at a concentration of 80 mg/mL. The mixture was then filtered using a sterile 100 µm cell strainer (Falcon) resulting in a uniform suspension of fecal slurry (FS). Aliquots of FS were stored at −80 °C, and each sample was thawed only once. An FS dose of 1.3 mg/g resulted in death in 70% of the inoculated animals and was therefore used in all experiments to induce sepsis.

All mice received either FS or an equivalent volume of vehicle (5% dextrose) via intraperitoneal injection. All mice were also administered subcutaneous injections of buprenorphine (0.05 mg/kg) in accordance with the Canadian Council on Animal Care’s guidelines for reducing or abolishing pain in animals. For the survival experiments, the mice were kept for 48 h or until humane endpoints were reached. For all other experiments, the surviving mice were anesthetized with isoflurane and euthanized via exsanguination and excision of the heart 18 h post-FS injection. Peritoneal lavage was then performed using ice-cold phosphate-buffered saline. The thoracoabdominal cavity was subsequently opened, and tissues were collected (snap-frozen in liquid nitrogen and stored at −80 °C). The serum levels of alanine aminotransferase (ALT), aspartate aminotransferase (AST), and lipase, as well as of creatinine and blood urea nitrogen (BUN), and albumin were measured to assess the level of systemic tissue damage via a Catalyst One Chemistry Analyzer (IDEXX Laboratories, Markham, ON, Canada).

### 2.3. Multiplex Cytokine and Tissue Analysis

The frozen tissues were thawed, weighed, and homogenized in a Tissue Protein Extraction Reagent (T-PER^TM^) containing Complete Mini Protease Inhibitor Cocktail (Complete™, Hoffman-La Roche Ltd. Mississauga, ON, Canada) at 4 °C. The homogenates were centrifuged at 9000× *g* for 10 min. The supernatants were collected and snap-frozen in liquid nitrogen for subsequent analysis. Cytokines IL-1β, IL-6, IL-10, TNFα, and keratinocyte chemoattractant/human growth-related oncogene (KC/GRO) levels in the serum and in lung and kidney homogenates, as well as in peritoneal lavage were measured via the V-plex mouse kit (Meso Scale Discovery) and normalized to protein content. Myeloperoxidase (MPO) activity was used as a marker for neutrophil content [21] and was measured in tissue homogenates using a colorimetric assay as previously described [22].

### 2.4. Peritoneal Bacterial Load

The peritoneal lavages were cultured on Centre for Disease Control blood agar plates (Anaerobe Systems) to assess bacterial clearance. The cultures were grown in anaerobic conditions for 24–48 h at 37 °C and subsequently counted; the results were expressed as colony-forming unit (CFU) × 10^9^/mL.

### 2.5. Gentamicin Protection Assay

Peritoneal macrophages from the mice were isolated as described [23] and cultured with *Escherichia coli* (*E. coli*) strain HB101 in Petri dishes at a multiplicity of infection of 10 *E. coli* per macrophage. The gentamicin protection assay was performed as described previously [24]. The data are presented as percent of bacteria killed compared to time 0.

### 2.6. Flow Cytometry Analysis

The macrophages isolated via peritoneal lavage (as described above) were fixed with ice-cold formalin. Phenotyping was performed as previously described [23]. Briefly, ~1.5 × 10^6^ cells fixed-cells were placed into round-bottom polypropylene tubes, blocked with 5% BSA for 30 min on ice. The fixed cells were then mixed with the antibodies anti-F4/80-PE/Cy5 at 0.2 mg/mL (eBioScience), anti-CD11b-PE at 1.6 mg/mL (Abcam), and anti-Ly6c-PE/Cy7 at 0.2 mg/mL (Abcam). The cells were gated for CD11b^+^ F4/80^+^ Ly6c^[low or high]+^.

### 2.7. Statistical Analyses

The data are presented as mean ± standard error (SEM) unless otherwise indicated. Student’s *t* test or 2-way analysis of variance with Sidak’s multiple comparison posthoc test was used to compare groups. Kaplan–Meier curves were obtained to depict survival; the survival analysis was performed using the Mantel Cox log rank test. All data analyses were performed using GraphPad Prism 9 (GraphPad Software, La Jolla, CA, USA). Statistical significance was set at *p* < 0.05.

## 3. Results

### 3.1. OSMR Type II Deficiency Improves Survival in Mice

The induction of FIP by intraperitoneal injection of FS decreased survival in aged WT and KO strains compared to vehicle-treated mice (Figure 1). Twenty-four hours after the injection of FS, 30% of WT mice survived, whereas 60% of KO mice survived; this corresponds to a 30% improved survival in aged KO mice compared to their WT counterparts. There were no differences in clinical score (data not shown) or survival rate between male and female mice; therefore, male and female mice were combined for the subsequent analyses.

### 3.2. OSMR Type II Deficiency Decreases Pro-Inflammatory and Anti-Inflammatory Responses in Mice

To study the pathophysiological mechanisms of sepsis, we endeavored to choose a time point in the progression of FIP associated with severe inflammation that preceded all deaths. In pilot experiments (using WT and KO mice at 12 weeks of age), we identified 18 h post-injection of FS as a time point associated with pronounced increases in systemic and tissue cytokines levels (data not shown). Moreover, pilot experiments revealed evidence of organ dysfunction (Figure 2), including elevated levels of BUN (Figure 2A), creatinine (Figure 2B), ALT (Figure 2C), as well as reduced levels of albumin (Figure 2D) in FIP mice compared to their vehicle-treated counterparts 18 h post-FS injection; however, there was no effect of the genotype on these parameters, despite young KO mice also showing improved survival compared to their WT counterparts (data not shown).

At 18 h post-injection of FS in aged mice, a pronounced elevation of IL-1β, IL-6, TNFα, and KC/GRO levels was seen in the serum samples of FIP mice compared to those of vehicle-treated mice (Figure 3A–E), suggesting systemic inflammation. There was an overall effect of the genotype on all cytokines analyzed, such that the KO mice had mitigated increases in serum pro- and anti-inflammatory cytokines, including IL-6 (Figure 3B), TNFα (Figure 3C), KC/GRO (Figure 3D), and IL-10 (Figure 3E). In contrast, there was no overall effect of the genotype on the serum IL-1β levels. We then examined the cytokine levels in various compartments, including peritoneal lavage samples (Figure 4), as well as in the lung, kidney, and liver (Table 1). Invariably, the cytokines levels were markedly elevated in FIP mice compared to vehicle-treated mice, as expected. The KO mice had attenuated inflammatory responses compared to the WT mice; this was evident in the peritoneal lavage cytokine levels (Figure 4A–C,E), except for the KC/GRO levels (Figure 4D). Similarly, the tissue levels of these cytokines in the lung and kidney were attenuated (Table 1). Interestingly, the cytokine levels in the liver exhibited different trends; whereas IL-10 was reduced in KO mice compared to WT mice in the liver, the levels of IL-6 and KC/GRO were unaffected by the genotype, and IL-1β and TNF were increased in KO compared to WT mice (Table 1).

To determine if the reduced cytokine levels coincided with reduced neutrophil infiltration, a myeloperoxidase (MPO) activity assay was performed in lung tissue. Eighteen hours post-FS injection, the FIP mice showed pronounced elevations in MPO activity, suggesting neutrophil infiltration, compared to vehicle-treated mice. Notably, the KO FIP mice had comparatively greater lung tissue infiltration compared to the WT FIP mice (Figure 5). MPO activity was also assessed in a limited number of peritoneal lavage samples, as well as in liver and kidney homogenates, but the assay signals were comparatively low and therefore not pursued in WT and KO FIP mice.

### 3.3. OSMR Type II Deficiency Affects Macrophage Function

We next examined macrophage populations in the peritoneum by flow cytometry. Notably, the flow cytometry protocol used herein did not distinguish between live and dead cells, to provide a complete picture of cells at the site of inoculation. The flow cytometry analysis revealed no changes in resident macrophages, gated as CD11b^+^ F4/80^+^ Ly6c^[low]+^, due to either FS or genotype (Figure 6A). Conversely, there was a greater infiltration of macrophages, gated as CD11b^+^ F4/80^+^ Ly6c^[high]+^, in the peritoneum of WT mice than in KO mice following injection of FS (Figure 6B). We next examined the bacterial killing efficiency of macrophages isolated from the peritoneum. No differences in bacterial uptake (at time 0) were evident between WT and KO FIP mice (data are mean ± SEM × 10^9^ CFU/mL: WT: 449 ± 26 (n = 3); Aged KO: 397 ± 25 (n = 4); *p* = 0.22). The gentamicin protection assay revealed that the KO mice had a higher *E. coli* killing capacity than their WT counterparts (Figure 6C). Finally, to assess the functional outcomes associated with these changes in macrophage function, we assessed the peritoneal bacterial loads 18 h post-injection of FS and found that the KO mice demonstrated worse peritoneal bacterial clearance than the WT mice (Figure 6D).

## 4. Discussion

In this study, we assessed the impact of OSMR type II signaling on survival and immune markers following induction of FIP in a murine OSMR type II KO model. Prior studies have shown that the OSM is increased in sepsis [8,19], and some studies suggest it plays both direct and indirect (i.e., via modulating immune responses) roles in inflammatory conditions. To summarize the findings of the present study, OSMR type II KO mice had (1) improved survival outcomes at 48 h; (2) reduced systemic and local inflammation 18 h after the induction of sepsis; and (3) altered macrophage function at the site of infection. Together, these data suggest that the OSM type II receptor signaling and/or its cognate ligands play a role in mediating the morbidity and mortality from sepsis in a murine model.

Sepsis was recently redefined by the joint task force of the Society of Critical Care Medicine and the European Society of Intensive Care Medicine as a “life-threatening organ dysfunction caused by a dysregulated host response to infection” [25]. This terminology update reflects a more nuanced appreciation of the role of immune dysregulation, involving both proinflammatory and immunosuppressive components within the pathophysiology of sepsis. Both pro- and anti-inflammatory profiles are associated with increased mortality from sepsis [26,27,28], suggesting an imbalance in immune function, rather than a specific effector, is associated with reduced survival.

The initial phase of peritonitis, which precedes the progressive and systemic deterioration of tissue function, was the primary focus of this study. Arguably, this phase would be most suitable for therapeutic intervention to prevent or mitigate sepsis-induced organ damage [2]. We found that FIP in WT mice caused widespread increases in both pro- and anti-inflammatory mediators (including IL-1β, IL-6, TNFα, KC, and IL-10), concomitant with increased infiltration of in CD11b^+^ F4/80^+^ Ly6c^[high]+^ cells—a subset of macrophages recruited to sites of infection where they phagocytose pathogens and induce local proinflammatory cytokine production [29,30,31,32]. In contrast, OSMR type II KO mice had less severe cytokine responses. Indeed, even at the site of infection, the cytokines levels were almost invariably reduced in the KO mice, and this coincided with reduced infiltration of macrophages in the peritoneum. The bactericidal actions of infiltrating leukocytes in response to infection are mediated by phagocytosis, the release of cytokines and proteases, and the generation of potentially cytotoxic levels of reactive oxygen species. The dysregulation of these processes, as occurs in sepsis, is thought to contribute to tissue injury and organ dysfunction. The impaired macrophage recruitment in OSMR type II KO mice may reduce systemic and local inflammation and tissue injury, resulting in improved survival in polymicrobial sepsis. Studies utilizing macrophage-specific OSMR type II deficiency could provide insights into these beneficial effects.

Alterations in innate immune responses may also affect the susceptibility to infection and sepsis [33,34,35]. Macrophages isolated from OSMR type II KO mice had superior bacterial killing capacity in vitro than those from WT controls, a factor that could contribute to the increased survival in this group. Notwithstanding, the higher bacterial loads in the peritoneum 18 h after the induction of sepsis suggest this increased bactericidal activity did not completely compensate for the reduced macrophage infiltration. Yet, the improved survival in this group, despite higher bacterial loads, further emphasizes the critical role of a dysregulated host response in the pathophysiology of sepsis and does not depend solely on the infectious organism per se.

There are contrasting views as to whether OSM constitutes a pro-inflammatory or an anti-inflammatory mediator. The administration of OSM has been shown to promote polymorphonuclear adhesion and transmigration into endothelial cells and to increase the expression of PMN activators (e.g., IL-6) [36]. In contrast, Wallace et al. showed that the in-vivo administration of OSM was associated with reduced inflammation and mortality in a model of acute inflammatory disease [37]. The findings from the present study suggest that OSM signaling via the OSMR type II receptor modulates both pro- and anti-inflammatory responses, and therefore its cumulative effects on the immune function are likely more nuanced and context specific. For example, our previous studies showed that genetic deletion of OSMR conferred protection in the lungs but caused more severe renal damage and increased mortality in a model of acute intestinal inflammatory disease [38]. These outcomes were attributed, at least in part, to a more pronounced anti-inflammatory response, characterized by increased IL-10 levels in KO mice, thus limiting their ability to mount an effective immune response. Here, after the induction of sepsis, IL-10 levels were attenuated in OSMR KO mice following the induction of FIP, as were the levels of the pro-inflammatory cytokines IL1β, IL-6, TNF, and KC, which may reflect a more balanced attenuation of inflammation with improved survival. Yet, while macrophage infiltration within the peritoneum was reduced, increased MPO activity in the lungs suggests increased neutrophil invasion within that tissue, supporting the notion that the effect of OSMR abrogation is likely cell- and tissue-specific and depends on physiological and pathophysiological circumstances in the host.

The benefits of an attenuated inflammatory response secondary to OSMR type II abrogation may extend beyond the immediate survival. Animal and human studies show that recovery from critical illness is associated with an increased risk of post-discharge morbidity and mortality from new-onset cardiovascular and renal disease [39], and the post-discharge risk tends to correlate with the severity of the critical illness. The prevailing notion is that immune, metabolic, and neurohumoral responses associated with critical illness induce a prolonged pro-inflammatory and immunosenescent state, resulting in cardiac and vascular remodeling and eventual cardiovascular dysfunction [39]. In this regard, the mitigated inflammatory response may in turn be associated with reduced post-illness complications. There are caveats. First, while inflammation is thought to be an important mediator, regional increases in lung MPO, which suggests increased regional neutrophil infiltration in some tissues, may exacerbate the damage in those tissues and thus offset the long-term survival advantage. Second, although the severity of illness was found to correlate with the post-illness risk of complications, the role of inflammation per se in dictating post-sepsis outcomes is difficult to study, particularly in studies involving humans. That is, with a mitigated inflammatory response, the severity of illness may be reduced in tandem, but the subsequent risk of long-term dysfunction remains elevated due to the unmitigated damage caused by the infectious agent. Indeed, the increased levels of BUN, creatinine, ALT, and LDH and the reduced levels of albumin indicate a profound multi-organ dysfunction in both WT and KO mice, which may be associated with long-term morbidity despite the increased short-term survival in the latter group. Future studies will be needed to resolve these issues.

As noted above, a majority of ICU admissions are patients over the age of 65 [40]. Despite this disproportionate representation of aged patients, the majority of preclinical studies use young rodents as model systems [41]. For this reason, we utilized an aging model of polymicrobial peritonitis. Although no comparisons were made with young mice, we found that aged WT and KO mice had reduced survival compared to their young counterparts in pilot studies (unpublished observations), which is consistent with other studies [42,43,44]. This outcome was expected, given that elderly patients not only are predisposed to sepsis (due to existing co-morbidities, increased hospitalizations, etc.), but also have poorer outcomes due to the age-related physiological decline (e.g., immunity, cardiovascular function, etc.) [45]. The increased susceptibility to sepsis in the elderly is attributed, at least in part, to dysregulated immune function [46]. Notably, older age is associated with chronic elevations in inflammatory cytokines [42,43,44], and higher levels of cytokines tend to be associated with poorer outcomes in sepsis [47]. Hence, decreasing the levels of circulating cytokines, via the inhibition of the OSM/OSMR type II pathway (which we attribute to reducing the circulating and tissue cytokine levels) or via alternative anti-inflammatory therapies, may be particularly beneficial in aged mice with chronic inflammation, and these beneficial effects may have been overlooked in young mice.

Although employing aging mice to model sepsis in humans is a strength of the study, the age range of the mice used was considerable (50–70 weeks old). With a typical lifespan of over 2 years, mice within this range are more likely to model humans of middle age, rather than the elderly that make a preponderance of septic patients. The choice to use advanced-age mice instead of elderly mice (>2 years) was a pragmatic one to minimize the attrition due to age-related illness. Nevertheless, the increasing age could differentially affect the immune responses in WT and KO mice in sepsis, and this should be examined in future studies. There are other limitations of this study that warrant discussion. Mice inoculated with FS were given neither antibiotics nor fluids, which constitute the standard of care for septic patients. Although this model does recapitulate a select group of cases in which sepsis goes undiagnosed and untreated (and thus represents its most severe form), the inclusion of standard of care would increase the translatability of this work. Additionally, tissues were collected at only a single time point in the progression of sepsis to interrogate the role of OSM signaling therein. Multiple time points would have provided more valuable insights, particularly since the transition from pro-inflammatory to immunosenescent phases in sepsis is thought to be important in dictating recovery and predicting long-term outcomes. The examination of the organ damage markers BUN and creatinine in aged mice would also provide additional insight into sepsis severity in these mice. Finally, sex differences are increasingly being recognized in relation to both the susceptibility and the pathophysiology of sepsis [48]. Here, male and females were combined, and though the results tended to be consistent among groups, future studies should endeavor to investigate the respective responses separately.

## 5. Conclusions

The findings presented herein demonstrate that the knockout of OSMR type II and therefore its signaling pathways improved survival in mice of advanced age. These results suggest that targeting this pathway may provide beneficial effects against the progression of sepsis clinically. The proposed mechanism involves a reduction in systemic pro-inflammatory and anti-inflammatory mediators and reduced macrophage migration to the site of infection. These findings also highlight the importance of including advanced-age models, especially in disease studies where elderly populations are particularly impacted. Understanding the mechanisms underlying the pathophysiology of sepsis, particularly in vulnerable populations like the elderly, may lead to the identification of novel intervention strategies [3].

## Figures and Tables

**Figure 1 biomedicines-11-00483-f001:**
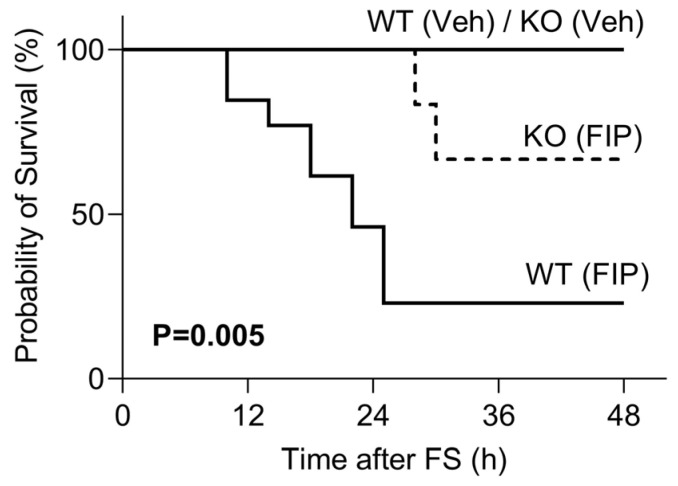
OSMR type II receptor knockout (KO) confers a survival advantage compared to wild-type (WT) mice subjected to fecal slurry-induced peritonitis (FIP). Kaplan–Meier curves depicting the survival of OSMR type II receptor knockout (KO) and FIP mice that received an intraperitoneal injection of 1.3 mg/g of fecal slurry (FS). WT Veh (n = 3); KO Veh (n = 3); WT FIP (n = 14); KO FIP (n = 6). *p* = 0.005 via log-rank (Mantel-Cox) test.

**Figure 2 biomedicines-11-00483-f002:**
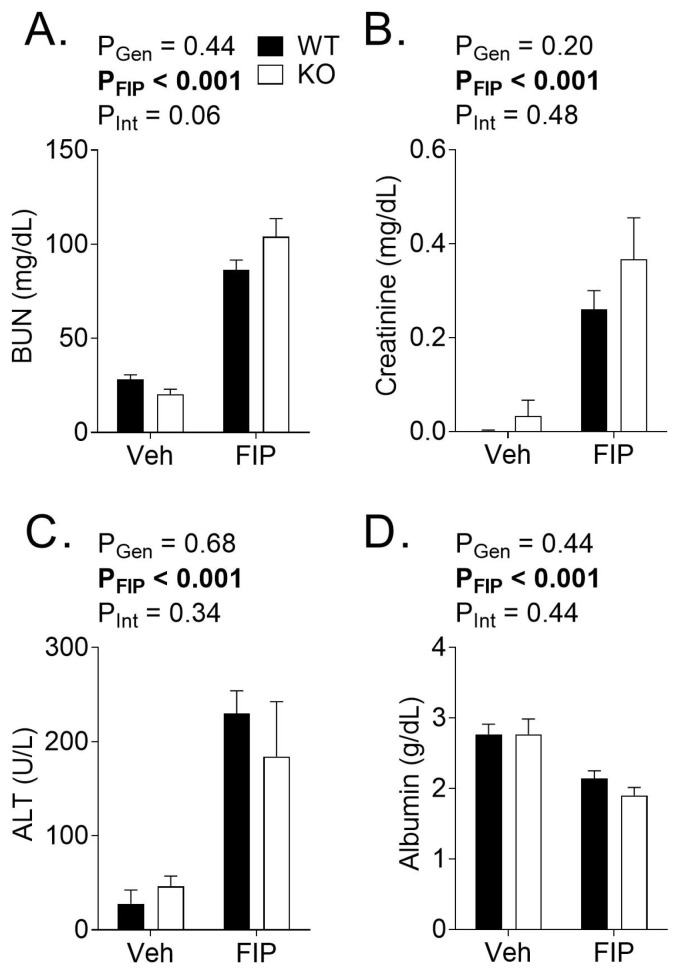
Serum biochemistry in wild-type (WT) and OSMR type II receptor knockout (KO) mice subjected to fecal slurry (FS)-induced peritonitis (FIP) or treated with vehicle (Veh). Serum collected 18 h post-injection of Veh or FS was analyzed for the levels of (**A**) blood urea nitrogen (BUN), (**B**) creatinine, (**C**) alanine aminotransferase (ALT), and (**D**) albumin. *p* values reflect the outcomes of 2-way analysis of variance. Each bar represents n = 3–5 mice.

**Figure 3 biomedicines-11-00483-f003:**
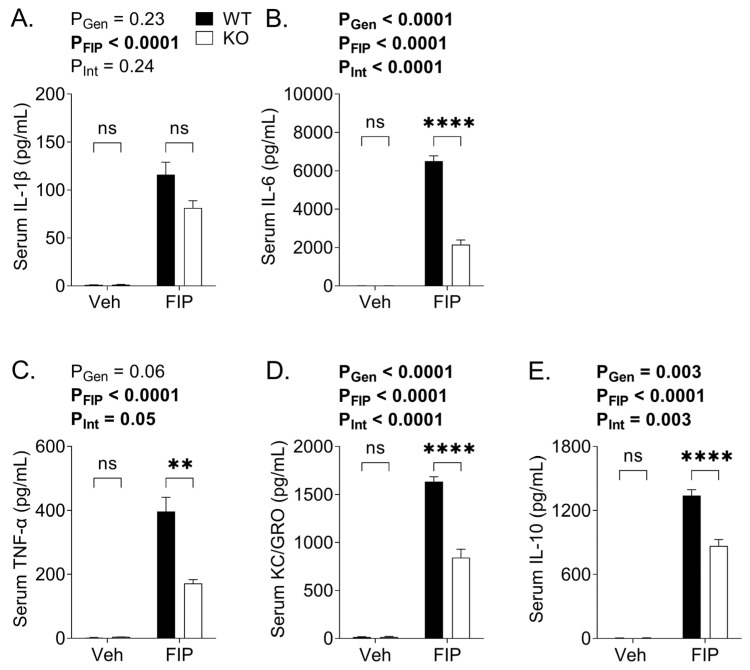
Serum cytokine profiles in wild-type (WT) and OSMR type II receptor knockout (KO) mice subjected to fecal slurry (FS)-induced peritonitis (FIP) or treated with vehicle (veh). Serum collected 18 h post-injection of Veh or FS was analyzed for the levels of (**A**) IL-1β, (**B**) IL-6, (**C**) tumor necrosis factor alpha (TNFα), (**D**) keratinocyte chemoattractant (KC)/human growth-related oncogene (GRO), and (**E**) IL-10. The *p* values reflect the outcomes of 2-way analysis of variance; Sidak post-hoc test outcomes are shown as ** *p* < 0.01, **** *p* < 0.0001, ns not significant. Each bar represents n = 3–13 mice.

**Figure 4 biomedicines-11-00483-f004:**
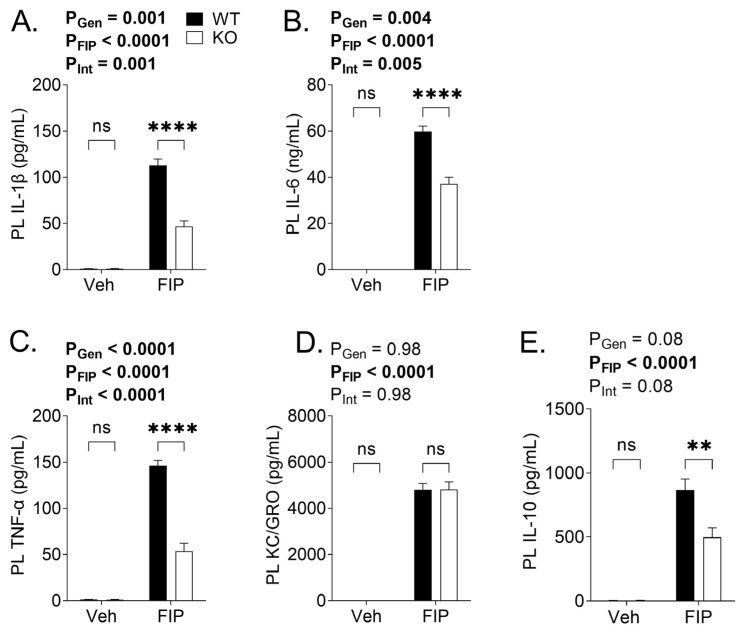
Peritoneal lavage (PL) cytokine profiles in wild-type (WT) and OSMR type II receptor knockout (KO) mice subjected to fecal slurry (FS)-induced peritonitis (FIP) or treated with vehicle (Veh). PL samples collected 18 h post-injection of Veh or FS were analyzed for the levels of (**A**) IL-1β, (**B**) IL-6, (**C**) tumor necrosis factor alpha (TNFα), (**D**) keratinocyte chemoattractant (KC)/human growth-related oncogene (GRO), and (**E**) IL-10. The *p* values reflect the outcomes of 2-way analysis of variance; Sidak post-hoc tests outcomes are shown as ** *p* < 0.01, **** *p* < 0.0001, ns not significant. Each bar represents n = 3–14 mice.

**Figure 5 biomedicines-11-00483-f005:**
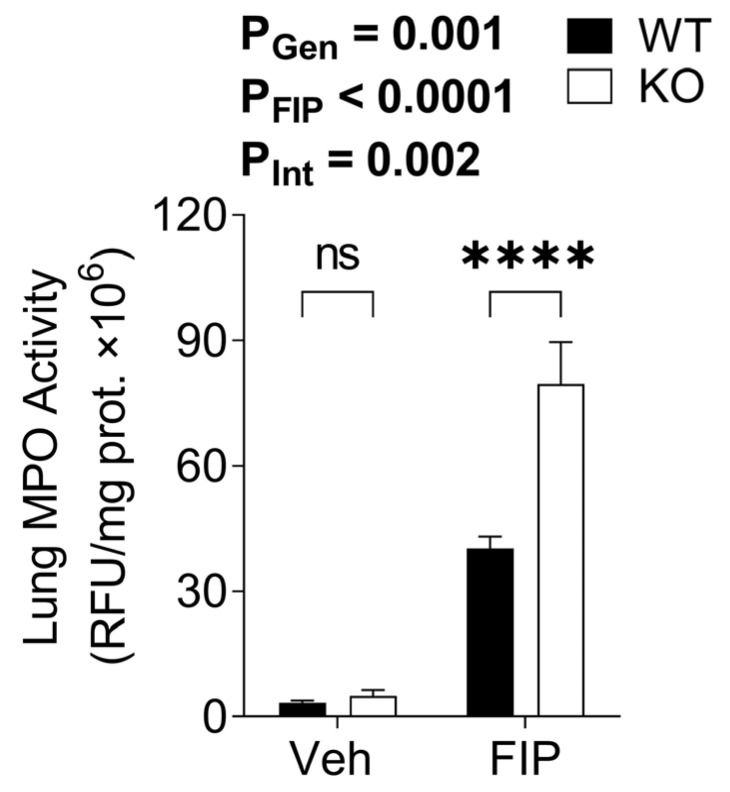
Lung neutrophil infiltration, as assessed by myeloperoxidase activity (MPO,) in wild-type (WT) and OSMR type II receptor knockout (KO) mice subjected to fecal slurry (FS)-induced peritonitis (FIP) or treated with vehicle (Veh). The *p* values reflect the outcomes of 2-way analysis of variance; Sidak post-hoc tests outcomes are shown as **** *p* < 0.0001, ns not significant. Each bar represents n = 3–12 mice.

**Figure 6 biomedicines-11-00483-f006:**
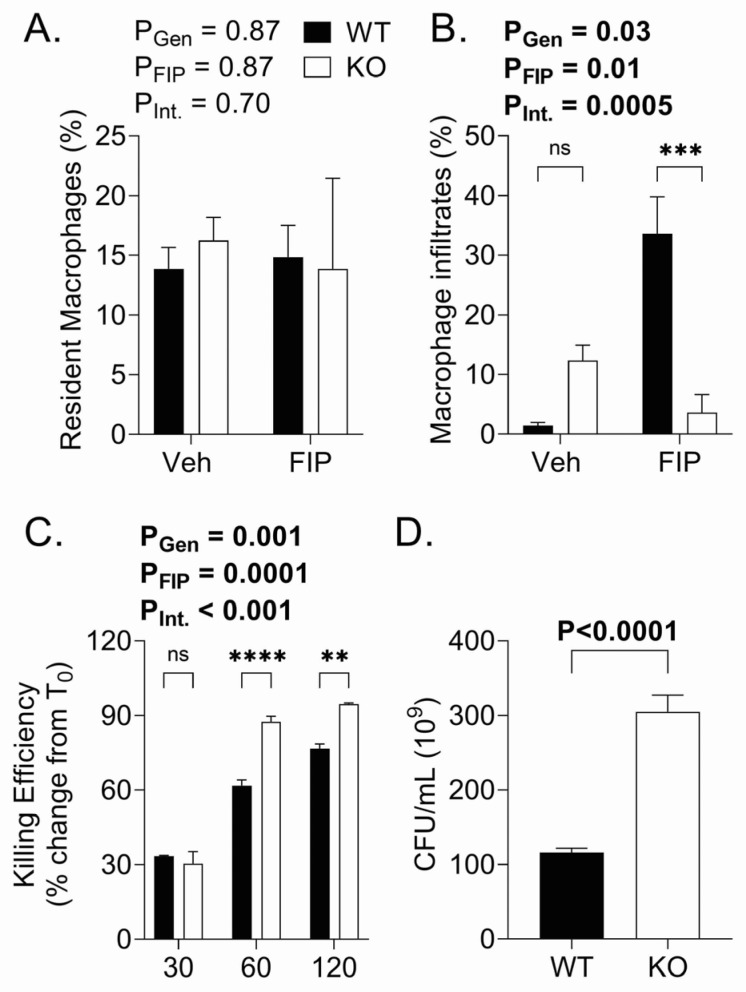
(**A**) Resident and (**B**) infiltrated macrophages assessed in peritoneal lavage in wild-type (WT) and OSMR type II receptor knockout (KO) mice subjected to fecal slurry-induced peritonitis (FIP) or injected with vehicle (Veh). The resident macrophages depicted in panel (**A**) were identified as CD11b+ F4/80+ Ly6c[low]+, and the macrophage infiltrates depicted in panel (**B**) were identified as CD11b+ F4/80+ Ly6c[high]+. (**C**) In vitro bacterial killing assay for wild-type (WT) and OSMR type II receptor knockout (KO) mice. (**D**) KO mice had reduced peritoneal bacterial clearance compared to WT mice. (**D**) In panels (**A**,**B**), the *p* values reflect the outcomes of 2-way analysis of variance; Sidak post-hoc tests outcomes are shown as ** *p* < 0.01, *** *p* < 0.001, **** *p* < 0.001, ns not significant. In panel (**D**) the, *p* values reflect the outcome of Student’s *t* test. Each bar represents n = 3–14 mice.

**Table 1 biomedicines-11-00483-t001:** Tissue cytokines/chemokine profiles.

	Veh		FIP		*p* Values		
Lung	WT	KO	WT	KO	Gen.	FIP	Int.
IL-1β (pg/mL)	2.7 ± 0.2 (3)	3.4 ± 0.3 (4)	178.5 ± 15.6 (12)	99.6 ± 2.3 (12) ****	**0.02**	**<0.0001**	**0.01**
IL-10 (pg/mL)	2.4 ± 0.1 (3)	2.6 ± 0.9 (4)	269.3 ± 12.1 (15)	167.7 ± 9.2 (7) ****	**0.005**	**<0.0001**	**0.005**
IL-6 (ng/mL)	16.4 ± 0.8 (3)	11.2 ± 5.1 (4)	3168.4 ± 162.4 (15)	1377.4 ± 152.0 (9) ****	**0.0005**	**<0.0001**	**0.0005**
TNF-α (pg/mL)	3.9 ± 1.1 (3)	4.1 ± 1.0 (4)	236.9 ± 8.8 (15)	150.6 ± 4.2 (8) ****	**0.0008**	**<0.0001**	**0.0008**
KC/GRO (pg/mL)	16.5 ± 4.3 (3)	7.5 ± 1.7 (4)	2668.9 ± 104.0 (14)	1491.2 ± 84.5 (8) ****	**0.001**	**<0.0001**	**0.001**
**Kidney**							
IL-10 (pg/mL)	4.1 ± 0.3 (3)	4.5 ± 0.3 (3)	209.5 ± 13.0 (10)	59.1 ± 5.1 (8) ****	**<0.0001**	**<0.0001**	**<0.0001**
IL-6 (ng/mL)	16.4 ± 0.8 (3)	27.1 ± 7.3 (3)	2388.6 ± 107.7 (11)	1339.3 ± 54.8 (6) ****	**0.0007**	**<0.0001**	**0.0006**
IL-1β (pg/mL)	2.2 ± 0.2 (3)	2.8 ± 0.8 (3)	239.3 ± 22.2 (12)	93.7 ± 11.0 (7) ****	**0.01**	**<0.0001**	**0.01**
TNF-α (pg/mL)	2.5 ± 0.5 (3)	3.8 ± 1.2 (4)	57.6 ± 3.2 (14)	43.9 ± 1.4 (4) *	**0.18**	**<0.0001**	**0.11**
KC/GRO (ng/mL)	6.3 ± 0.5 (5)	10.4 ± 3.1 (3)	911.0 ± 81.5 (15)	488.9 ± 45.1 (6) **	**0.06**	**<0.0001**	**0.05**
**Liver**							
IL-10 (pg/mL)	2.2 ± 0.7 (3)	1.4 ± 0.2 (3)	459.4 ± 80.9 (5)	152.2 ± 16.4 (6) ***	**0.01**	**<0.0001**	**0.01**
IL-6 (ng/mL)	0.01 ± 0.0 (3)	0.03 ± 0.0 (3)	13.76 ± 1.41 (12)	10.03 ± 1.12 (10)	0.67	**<0.0001**	0.66
IL-1β (pg/mL)	2.6 ± 1.6 (3)	15.5 ± 4.1 (3)	870.1 ± 90.1 (9)	1541.9 ± 224.7 (4) **	**0.03**	**<0.0001**	**0.04**
TNF-α (pg/mL)	1.5 ± 0.4 (3)	2.0 ± 0.5 (3)	65.6 ± 8.2 (13)	124.9 ± 16.4 (6) **	**0.05**	**<0.0001**	**0.05**
KC/GRO (pg/mL)	4.9 ± 2.3 (3)	7.8 ± 2.7 (3)	3450.7 ± 273.4 (12)	3482.7 ± 261.5 (4)	0.97	**<0.0001**	0.96

Tissues were collected from mice 18 h after FS injection. The *p* values reflect the outcomes of 2-way analysis of variance; Sidak post-hoc test outcomes are shown as * *p* < 0.05, ** *p* < 0.01, **** *p* < 0.0001. Data are mean ± SEM (n).

## Data Availability

Not applicable.

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
