# Peer review of "Oncostatin M Receptor Type II Knockout Mitigates Inflammation and Improves Survival from Sepsis in Mice"

_biomedicines, 2023, doi:10.3390/biomedicines11020483_

Round 1

Reviewer 1 Report

1) Abstract.This coincided with reduced systemic pro-inflammatory and anti-inflammatory markers, increased bactericidal properties of macro-phages, and reduced macrophage migration to site of infection. Please, improve the description of conclusions.

2) 1. Introduction L32-40. Sepsis is a life-threatening organ dysfunction caused by a dysregulated host response  to a pathogen. Despite advances in critical care, sepsis is associated with high mortality  rates [1-3], and remain leading causes of death of hospitalized patients, especially in the  Intensive Care Unit [4, 5]. Cytokines are key players in the development of sepsis and  septic shock and profiling of septic patients has demonstrated increases in multiple pro-  and anti-inflammatory cytokines, including Tumor Necrosis Factor α (TNFα), interleukin 38 (IL)-1β, IL-6, IL-8, and IL-10 [6, 7]. I believe that authors need to enrich their manuscript with more citations from updated and recent paper, such as: 

a- Cytokine Profiles as Potential Prognostic and Therapeutic Markers in SARS-CoV-2-Induced ARDS. J Clin Med. 2022;11(11):2951. Published 2022 May 24. doi:10.3390/jcm11112951

b- Cytokine elevation in severe and critical COVID-19: A rapid systematic review, meta-analysis, and comparison with other inflammatory syndromes. Lancet Respir. Med. 2020;8:1233–1244. doi: 10.1016/S2213-2600(20)30404-5.

3) Introduction. L67-71. Although individuals of all ages can develop sepsis and septic shock, the elderly (>65  years of age) account for a preponderance of affected patients [18]. Advanced age is  known to be an independent risk factor for sepsis-related mortality, and yet elderly pa-  tients are often excluded from studies on the diagnosis and management of sepsis [19]. As  the elderly population grows, the use of aging models of disease is becoming increasingly  timely and relevant. Therefore, the experimental design used mice of advanced age. Please, improve this paragraph.

4) 3. Results L145-148. 3.1. OSMR type II deficiency improves survival in mice  Induction of FIP by intraperitoneal injection of FS decreased survival in WT and KO  strains compared to vehicle-treated mice (Figure 1). Please underline in the paper the most important statistical values.

5) 4. Discussion L245-247. Sepsis was recently redefined by the joint task force of the Society of Critical Care  Medicine and the European Society of Intensive Care Medicine as a “life-threatening or-  gan dysfunction caused by a dysregulated host response to infection” [25]. Please, summarise here the most important results of the study.

6) 5. Conclusions L369-374. The findings presented herein suggest that targeting the OSM/OSMR type II pathway  may provide beneficial effects against the progression of sepsis. The proposed mechanism  involves a reduction in systemic pro-inflammatory and anti-inflammatory markers and a  reduction of inflammatory macrophage migration to the site of infection. Understanding  the mechanisms underlying the pathophysiology of sepsis, particularly in vulnerable pop- ulation like the elderly, may lead to the identification of novel intervention strategies [3]. Please, improve the decsription of conclusions and underline the novelty of the study.

Author Response

Response: We thank the editor and reviewers for their constructive feedback. We have addressed all the comments and hope the reviewers find this revised manuscript suitable for publication.

1) Abstract - This coincided with reduced systemic pro-inflammatory and anti-inflammatory markers, increased bactericidal properties of macro-phages, and reduced macrophage migration to site of infection. Please, improve the description of conclusions.

Response: We have improved the abstracts description of conclusions, as suggested (lines 26-28).

2) 1. Introduction L32-40. Sepsis is a life-threatening organ dysfunction caused by a dysregulated host response to a pathogen. Despite advances in critical care, sepsis is associated with high mortality rates [1-3], and remain leading causes of death of hospitalized patients, especially in the Intensive Care Unit [4, 5]. Cytokines are key players in the development of sepsis and septic shock and profiling of septic patients has demonstrated increases in multiple pro- and anti-inflammatory cytokines, including Tumor Necrosis Factor α (TNFα), interleukin 38 (IL)-1β, IL-6, IL-8, and IL-10 [6, 7]. I believe that authors need to enrich their manuscript with more citations from updated and recent paper, such as: a-Cytokine Profiles as Potential Prognostic and Therapeutic Markers in SARS-CoV-2-Induced ARDS.J Clin Med. 2022;11(11):2951. Published 2022 May 24. doi:10.3390/jcm11112951; b-Cytokine elevation in severe and critical COVID-19: A rapid systematic review, meta-analysis, and comparison with other inflammatory syndromes. Lancet Respir. Med. 2020;8:1233–1244. doi: 10.1016/S2213-2600(20)30404-5.

Response: Thank you for the suggestion. We have updated the references and included the second reference (line 48); we chose to omit the first suggested reference because that one made no specific reference to patients with sepsis or septic shock.

3) Introduction. L67-71. Although individuals of all ages can develop sepsis and septic shock, the elderly (>65 years of age) account for a preponderance of affected patients [18]. Advanced age is known to be an independent risk factor for sepsis-related mortality, and yet elderly pa- tients are often excluded from studies on the diagnosis and management of sepsis [19]. As the elderly population grows, the use of aging models of disease is becoming increasingly timely and relevant. Therefore, the experimental design used mice of advanced age. Please, improve this paragraph.

Response: We have revised the paragraph and restructured the introduction to improve flow and clarity, as suggested (lines 34-44; 76-79).

4) 3. Results L145-148. 3.1. OSMR type II deficiency improves survival in mice Induction of FIP by intraperitoneal injection of FS decreased survival in WT and KO strains compared to vehicle-treated mice (Figure 1). Please underline in the paper the most important statistical values.

Response: It wasn’t altogether clear what the reviewer was asking here: to specifically “underline” or simply include p values in the text. If the reviewer means the latter, we felt the most important p values are clearly indicated in the figures. Moreover in many cases, a specific outcome cannot be summarized with a single p value, since many outcome depends on both the overall P values from the 2-way ANOVA, as well as the post hoc test. Therefore we would kindly ask the editors for guidance on this point.

5) 4. Discussion L245-247. Sepsis was recently redefined by the joint task force of the Society of Critical Care Medicine and the European Society of Intensive Care Medicine as a “life-threatening or-gan dysfunction caused by a dysregulated host response to infection” [25]. Please, summarise here the most important results of the study.

Response: We have reorganized the discussion so as to summarize the main findings of the study in the first paragraph of the discussion, as suggested (Lines 272-281).

6) 5. Conclusions L369-374. The findings presented herein suggest that targeting the OSM/OSMR type II pathway may provide beneficial effects against the progression of sepsis. The proposed mechanism involves a reduction in systemic pro-inflammatory and anti-inflammatory markers and a reduction of inflammatory macrophage migration to the site of infection. Understanding the mechanisms underlying the pathophysiology of sepsis, particularly in vulnerable pop- ulation like the elderly, may lead to the identification of novel intervention strategies [3]. Please, improve the description of conclusions and underline the novelty of the study.

Response: We have improved this section and reemphasized the novelty, as suggested (Lines 399-408).

Reviewer 2 Report

The manuscript by Salim et al. is well-written and the results are presented clearly. The manuscript has merit for publication in this journal, but there are some minor issues that need to be addressed before publication.

Minor comments

Line 92... the authors have mentioned that pathogen-free C57BL/6 mice were used for inducing sepsis. I wonder how pathogen-free cecal contents would induce sepsis.

Line 99… Please clarify whether buprenorphine was administered to control vehicle animals as well. Vehicle control mice should also receive Buprenorphine.

Line. 132… I don’t understand why fixation was performed before surface staining. That meant the authors have not included live/dead count dye in their analysis. Please include supplementary data on how live/dead gating was performed.

Line 147… I didn’t understand how the authors calculated a 60% survival rate. Please mention how many mice were dead and live at the end of the experiment.

Author Response

Response: We thank the editor and reviewers for their constructive feedback. We have addressed all the comments and hope the reviewers find this revised manuscript suitable for publication.

The manuscript by Salim et al. is well-written and the results are presented clearly. The manuscript has merit for publication in this journal, but there are some minor issues that need to be addressed before publication.

Minor comments
Line 92... the authors have mentioned that pathogen-free C57BL/6 mice were used for inducing sepsis. I wonder how pathogen-free cecal contents would induce sepsis.

Response: Good point! In this case, ‘pathogen-free’ was intended to mean that mice used to extract cecal contents had no evidence of pre-existing or underlying pathology. Infections are monitored in our animal care facility using a sentinel program. This allowed extraction of bacteria that exists typically within murine cecal contents. We have revised the methods section as follows: “Cecal contents from male and female C57BL/6 mice (9-13 weeks of age) with no signs of infection were collected,…” “ (lines 101-102).

Line 99… Please clarify whether buprenorphine was administered to control vehicle animals as well. Vehicle control mice should also receive Buprenorphine.

Response: The reviewer is correct in that control mice also received buprenorphine in our study. This has now been included in the manuscript (lines 108-113).

Line. 132… I don’t understand why fixation was performed before surface staining. That meant the authors have not included live/dead count dye in their analysis. Please include supplementary data on how live/dead gating was performed.

Response: The reviewer is correct that the cells for flow cytometry were fixed prior to staining. Prior to these experiments, we discussed the advantages and disadvantages to this approach, and ultimately settled on prior fixation for two reasons. First, the goal of this assay is to gain an understanding of how OSMR gene deletion affects macrophage infiltration in the peritoneal cavity; we reasoned that total macrophage infiltration should include analysis of both live and dead cells, since a large proportion of these cells are likely to die at the site of infection, and thus their exclusion would not give a full picture. Second, from a logistical perspective, prior fixation meant that cells could be analyzed at a convenient time without degradation of samples. The experiments were such that multiple mice needed to be euthanized at various time points throughout the day, and these days were often extremely busy. Fixation allowed for rapid processing, and then flow analysis could be performed when all samples had been collected. This was deemed the best approach, particularly since the 18h post-injection time point made for very long days (since monitoring had to be performed every 2-4 hours in the mice) throughout the duration of the experiment. Finally, it is also important to note that all samples from all groups were treated exactly the same, and thus any confounding effects of this approach in one group would presumably affect all groups, and therefore we are confident the results are robust. Notwithstanding, we have specified this detail in the results section so as to prevent any misunderstandings (lines 247-249).

Line 147… I didn’t understand how the authors calculated a 60% survival rate. Please mention how many mice were dead and live at the end of the experiment.

Response: The quoted survival rate reflects only the calculation made in the group of mice that were left indefinitely; none of these mice were euthanized for experimental end points. In other subgroups of animals, mice were euthanized specifically at 18 hours post injection of fecal slurry to study pathophysiological mechanisms in tissues; from these cohorts, survival could not be determined. This has been clarified in the methods (lines 108-113).
